# Epidemiological Analyses of the First Incursion of the Epizootic Hemorrhagic Disease Virus Serotype 8 in Tunisia, 2021–2022

**DOI:** 10.3390/v16030362

**Published:** 2024-02-27

**Authors:** Thameur Ben Hassine, José-María García-Carrasco, Soufien Sghaier, Sarah Thabet, Alessio Lorusso, Giovanni Savini, Salah Hammami

**Affiliations:** 1General Directorate of Veterinary Services, Regional Commissary for Agricultural Development of Nabeul, Nabeul 8000, Tunisia; 2Biogeography, Diversity and Conservation Lab, Department of Animal Biology, Faculty of Sciences, University of Malaga, E-29071 Malaga, Spain or jmgc@uma.es; 3Food and Agriculture Organisation (FAO), Subregional Office for North Africa, les Berges du Lac 1, Tunis 1053, Tunisia; sghaiersoufien@yahoo.fr; 4Institut de la RechercheVétérinaire de Tunisie, Tunis 1006, Tunisia; sarahthabetbenabdeljelil@gmail.com; 5Istituto Zooprofilattico Sperimentale dell’Abruzzo e del Molise, 64100 Teramo, Italy; a.lorusso@izs.it (A.L.); g.savini@izs.it (G.S.); 6École Nationale de Médecine Vétérinaire de Sidi Thabet (ENMV), Service de Microbiologie, Immunologie et Pathologie Générale, Université de la Manouba, Tunis 2020, Tunisia; saleehhammami@yahoo.fr

**Keywords:** EHDV-8, cattle, *Culicoides*, risk maps, Tunisia

## Abstract

Epizootic hemorrhagic disease (EHD) is a non-contagious arthropod-transmitted viral disease and a World Organization for Animal Health (WOAH)-listed disease of domestic and wild ruminants since 2008. EHDV is transmitted among susceptible animals by a few species of midges of genus *Culicoides*. During the fall of 2021, a large outbreak caused by the epizootic hemorrhagic disease virus (EHDV), identified as serotype 8, was reported in Tunisian dairy and beef farms with Bluetongue virus (BTV)-like clinical signs. The disease was detected later in the south of Italy, in Spain, in Portugal and, more recently, in France, where it caused severe infections in cattle. This was the first evidence of EHDV-8 circulation outside Australia since 1982. In this study, we analyzed the epidemiological situation of the 2021–2022 EHDV outbreaks reported in Tunisia, providing a detailed description of the spatiotemporal evolution of the disease. We attempted to identify the eco-climatic factors associated with infected areas using generalized linear models (GLMs). Our results demonstrated that environmental factors mostly associated with the presence of *C. imicola*, such as digital elevation model (DEM), slope, normalized difference vegetation index (NDVI), and night-time land surface temperature (NLST)) were by far the most explanatory variables for EHD repartition cases in Tunisia that may have consequences in neighboring countries, both in Africa and Europe through the spread of infected vectors. The risk maps elaborated could be useful for disease control and prevention strategies.

## 1. Introduction

Epizootic hemorrhagic disease virus (EHDV), which belongs to the genus *Orbivirus* and the Sedoreoviridae family, is a non-enveloped, double-stranded segmented RNA virus genome approximately 19–20 kb in length [1]. Since 1965, EHDV has attracted researchers’ attention through an outbreak in New Jersey, USA, which caused the death of more than 500 deer [2]. Since then, the virus has also been detected in Canada and Mexico, as well as in other parts of the world: South America, Africa, the Middle East, Japan, Southeast Asia, and Australia [3]. To date, seven serotypes of EHDV have been reported, named 1–2 and 4–8, which were identified based on phylogenetic studies, sequencing data, and cross-neutralization assays [4]. Genetic analyses demonstrated that the previously identified serotype 3 (Nigerian strain IbAr 22619) was serotype 1 [5].

The first EHDV serotype 6 outbreak in Tunisia was described in 2006 and was closely related to other EHDV-6 strains circulating in the Mediterranean basin during this period in Turkey, Morocco, Algeria, and Jordan [6]. The EHDV-7 strain was involved in outbreaks in Israel in 2006 [7]. A common “African/Arabian Peninsula and Indian Ocean Asia” origin of EHDV-6 and 7 is revealed by analysis of their genomic segments [7]. Serological investigations revealed the circulation of EHDV-6 in Tunisia in 2012–2013 without reported clinical signs [8]. In 2015, few cattle cases of EHDV-6 were reported to the World Organization for Animal Health (WOAH) by Tunisian authorities [9]. During September 2016–February 2017, EHDV-1 was recorded in ruminants in the Middle East [10]. Phylogenetic analyses indicated a close relationship with the EHDV serotype 1 strain in Nigeria [7].

From September to November 2021, many clinical cases with BTV-like clinical signs were reported in Tunisian dairy and beef farms. At the beginning of the outbreak, Bluetongue virus serotype 4 (BTV-4) was suspected since it was detected in cattle in Tunisia in 2020 [11]. It was laterassociated with EHDV serotype 8. The virus was responsible for many clinical cases in cattle and a few deer deaths [12,13]. In 2022, a second EHDV epidemic was described in Tunisia between July and November. On 25 October 2022, EHDV first appeared on the European continent in Sicily and south-eastern Sardinia-Italy, causing BTV-like clinical symptoms in cattle [14]. On 18 November 2022, EHDV serotype 8 was identified as the etiological agent of a series of outbreaks that were discovered in southern Spain [14]. Furthermore, in July 2023, EHDV was reported for the first time in Portugal [8]. More recently, EHDV outbreaks were confirmed on 18 September 2023 in south-western France, close to the Spanish border (source: https://wahis.woah.org/; last accessed on 13 December 2023).

Available data in the literature suggest that the species of *Culicoides* involved in EHDV transmission are likely similar to those that transmit BTV [15,16]. Recent studies showed that EHDV-8 seems to use the same transmission patterns as BTV [17]. This in turn means that EHDV-8 has the potential to spread in Europe. Like other vector-borne diseases (VBDs), EHDV infections are typically seasonal and occur when vector insect populations are most abundant, usually from mid-summer to late autumn [18,19,20]. Eco-climatic factors are known to influence the distribution of competent insect vectors. Many researchers have evaluated the correlation between meteorological factors and the distribution of Bluetongue disease [11,21]. However, there are still gaps in our understanding of infection with EHDV, which impedes its control, particularly regarding the eco-climatic factors associated with virus circulation. In the present study, we investigated the effect of environmental and climatic drivers on this disease epidemic to provide accurate information for designing more effective surveillance and control systems.

## 2. Materials and Methods

### 2.1. Outbreak Dataset and Cluster Analysis

Available data of reported EHDV outbreaks in Tunisia during 2021 and 2022 were collected from the National Veterinary Research Institute (IRVT) of Tunisia. Only cases confirmed by serum-neutralization (SN) test and real-time RT-PCR were considered in this study. The methodology of confirmation was well described by Sghaier et al., in 2023 [12].

The spatiotemporal distribution of infection with EHDV is described. Spatial autocorrelation of EHDV cases was examined using the second- and third-level administrative division (Delegation and Imada). Imada is lowest-level administrative division in Tunisia. This enabled assessment of the importance of location in exposure to the virus. Cluster analysis was performed by calculating the local Moran’s indicator for the spatial autocorrelation (LISA) statistic [22]. Spatially based weighing and calculation of the LISA statistic and the global Moran’s I statistic were performed using Geoda software. The *p* value was then calculated through Monte Carlo hypothesis testing by comparing the rank (R) of the maximum likelihood from the real dataset with the maximum likelihoods from 999 random datasets.

### 2.2. Environmental Risk Models

First, we constructed a distribution model for the possible primary vector responsible for transmitting the EHDV, *C. imicola*, using occurrence points collected between 2017 and 2020 in Tunisia (64 presence points). The *C. imicola_F* variable goes beyond the vector’s own distribution, which may be greatly underestimated in the country, as it reflects environmentally favorable areas for the vector in Tunisia. This variable will allow the incorporation of information about the potential vector’s distribution into the EHDV model. This model was developed using the multGLM function in the fuzzySimR package v.4.9.9 package [23]. We used variables, such as topography, climate, water availability, livestock, anthropic, and vegetation (Table 1), that could be linked with the more proximal causal factors of the distribution of the vector [24]. The multicollinearity of the variables was controlled through Spearman correlation analysis. Spearman correlation analysis was used to identify pairs of variables with values above 0.8. If two variables had a correlation higher than 0.8 [25], the least informative variable with the distribution of the presence of *C. imicola* was excluded. The remaining variables were selected according to a forwardstepwise procedure based on the significance (*p* value) of the statistical tests [23].

The result generated by the GLM is a probability value related to the occurrences in the study area. This probability is influenced by the species’ response to the predictors as well as the species’ overall prevalence, where prevalence is the ratio of Imadas with presences to the total number of Imadas in Tunisia. To mitigate the impact of prevalence on the model’s output, we employed the favorability function [26]. For this purpose, the Fav function in fuzzySim uses the proportion of presence included in the model to calculate favorability, which reflects the degree (between 0 and 1) to which the local probability values differ from those expected according to the species prevalence.

Once we identified the most favorable areas for the distribution of *C. imicola* (*C. imicola_F*), we developed a model to identify the zones most prone to the circulation of EHDV. The procedure followed was the same and involved multicollinearity assessment (>0.8) and stepwise forward selection based on the *p* value. However, in this case, in addition to the variables shown in Table 1, the favorability of encountering *C. imicola* (*C. imicola_F*) was used as an additional variable for virus circulation. This variable is relevant for detecting favorable areas where the virus circulates, as it is considered the primary disease vector. We did not impose the use of the *C. imicola_F* variable on the model but allowed the model to analyze and include it along with the other variables if it were statistically related to the 2021 outbreaks. Therefore, this variable, together with the other variables (Table 1), underwent the forwardstepwise procedure. The resulting model’s probability was transformed into favorability, indicating the extent to which environmental conditions (and the distribution of *C. imicola*) can favor the occurrence of infection with EHDV cases, even in areas where it has not yet been reported. Therefore, favorability values were used as an indicator of risk.

### 2.3. Evaluation of the Model

We assessed model performance in terms of classification, discrimination capacity and calibration using the modEvA R package v. 3.9.3 [23]. The sensitivity, specificity, underprediction, overprediction, kappa [27], and true skill statistic (TSS) [28] were calculated using prevalence as the classification threshold. The overall ability of the model to discriminate between variables was evaluated by calculating the area under the curve (AUC) on the receiver operating characteristic (ROC) plot. The AUC provides a measure of the model’s ability to distinguish effectively across the full spectrum of prediction thresholds. We assessed the calibration of the models using the Hosmer and Lemeshow test [29], which is frequently used in risk prediction models. Models are considered well-calibrated if the observed and predicted rates are not significantly different. On the other hand, to assess the model’s ability to predict cases in 2022, we employed Miller’s calibration, which evaluates the bias and spread of predictions through its intercept and slope [30].

## 3. Results

### 3.1. Spatiotemporal and Cluster Analysis

In 2021, there were 161 total confirmed cases of EHDV: 5 (3%) were reported in late September, 125 (78%) were reported in October, and 31 (19%) were reported in November. The first three affected herds were reported on 28 September 2021 in the Kasserine (delegation of Sbitla) and Kairouan (delegation of HajebLayoun) governorates. The infection spread in the first half of October to central and north-western Tunisia (Kef, Jendouba, Siliana, Sidi Bouzid, and Gafsa governorates). In the second half of October, the disease was confirmed in the coastal region and in the northeast region of Tunisia. By the end of October, three cattle were confirmed to be EHDV positive in the oases of Tozeur. In November, the number of new cases declined in most regions of Tunisia. However, newly affected herds continued to be reported in the north, and in oases in the south (Kebili, Gabes, Tataouine, and Tozeur) (Figure 1).

In 2022, the disease appeared earlier. The first confirmed case was reported by the end of July (29 July 2022) in the governorate of Jendouba (Balta-BouAouane delegation). The disease spread to neighboring governorates in the north-west (Kef, Beja, Silian). The most important number of cases was recorded in September, especially in the north-east, Cap Bon, the center, and the region of Sahel of Tunisia. In October, the first cases were recorded in the southern region (in the oases of Tozeur and Kebili). Like in 2021, the number of new cases declined significantly in November, and the last case was reported on 23 November 2022. In total, 141 cases were confirmed in 1 July 2022, 35 in August (25%), 76 in September (54%), 23 in October (16%), and 6 in November (4%) (Figure 1).

As shown in Figure 1, the distributions of EHDV-infected herds in 2021 and 2022 were highest in central-west and north-western Tunisia. A high number of infected herds wasalso observed in the region of CapBon (north-east), as well as in the coastal plain of Tunisia. In the same governorate, infection was limited in some delegations (second-level administrative divisions).

Moran’s I statistics for EHD cases in 2021 and 2022 were 0.14 and 0.15, respectively, which were significantly greater than 0 (*p* < 0.0001 for both), indicating statistically significant spatial autocorrelation. The significant clusters are shown in Figure 2.

### 3.2. Environmental Risk Models

The presence of the *C. imicola* vector was favored by warm zones (positive NLST) with water availability, either from irrigated areas or areas near rivers. Additionally, it was favored by rural areas (with low population density) with densities of sheep livestock (Figure 3A). On the other hand, cases of EHDV occurred in areas that met these criteria for vector presence, as the inclusion of the vector variable (*C. imicola_F*) was the most influential variable (Table 2). Furthermore, these areas exhibited high variability in both temperature (lst_dn_diff) and vegetation (NVDI_diff). Among the zones favorable for vector circulation, those located inland (Figure 3B) demonstrated heightened variability, which was more favorable for EHDV cases. Furthermore, the most favorable areas were found to the north and center, away from arid areas and closer to the EHDV serotype 8 cases that Italy suffered in 2022 (Figure 4).

The models were well supported based on their evaluation. Both the *C. imicola* and the EHDV risk models showed high sensitivity and specificity and a low prediction rate (Table 3). Both models indicated acceptable discrimination capacities according to Hosmer and Lemeshow [31]. Differences between the expected and observed cases were not significant for either model (*C. imicola* HL = 8.59, *p* > 0.05; EHDV risk HL = 9.36, *p* > 0.05), indicating good calibration.

In 2021, there were 97 Imadas with EHDV cases; however, the next year, in 2022, 90 Imadas were positives. We assessed the ability of the EHDV risk model to predict cases for the following year; i.e., 2022. In 2022, 53% of the EHDV cases occurred in areas with F values >0.5, and 92% occurred in areas at risk (F > 0.2).The model constructed with 2021 infection with EHDV cases was calibrated using Miller’s calibration model with 2022 EHDV cases to determine how accurately it could predict cases. The Miller calibration line for 2022 had a slope of 0.443 and an intercept of −1.591. An intercept lower than 0 indicates that the model could not predict a decrease in the overall probability of occurrence. The slopes (0.443) and <1 suggest that predicted values below 0.5 underestimate the probability, whereas values above 0.5 overestimate the probability.

## 4. Discussion

Causing several clinical cases, EHDV can be a serious problem for herd cattle in the near future. Little is known about EHDV risk factor drivers, which can have implications for designing control strategies. As with other vector-borne diseases (VBDs), the distribution of EHD largely depends on the environmental factors that determine the abundance of the arthropod vector [32]. Ecological niche modeling (ENM) can be used to predict the abundance and the spread of VBDs [33] which in turn could be useful for planners in creating mosquito/VBD surveillance programs. A commonly used method to buildspecies distribution models (SDMs)in the ecological niche theory framework includes regression-based methods, such asgeneralized linear models (GLMs).This algorithm has widely proven to produce robust models for predicting species distribution [34,35,36,37,38] as well as for infectious diseases [39,40]. In this study, we elaborated distribution models (through GLMs) in order to: (1) establish the potential area of distribution of the potential vector in Tunisia, (2) to define the potential area of EHDV distribution cases. *C. imicola*, the main vector for Bluetongue in Tunisia, is a competent vector for EHDV in different parts of the world [16].Taking into account its high abundance, there is a high probability that *C. imicola* is the vector for EHDV in Tunisia. The favorability map of this vector was generated and subsequently, used as a covariate for infection with EHDV risk mapping. For the first step, we used *C. imicola* occurrence points collected between 2017 and 2020 (64 presence points). Predictors, biotic and abiotic rasters, were chosen on the basis of countrywide availability and associations already proven with the vector and the disease (topography, climate, water availability, livestock, anthropic, and vegetation). Using GLMs, we created a map that predicted the distribution of *C. imicola* throughout the contiguous region of Tunisia. The most important variables identified by the *C. imicola* model were related to temperature (positive night land surface temperature (NLST)) and irrigated area. Additionally, it was favored by rural areas (those with low population density) and those with densities of sheep livestock. The NLST has been identified as one of the most important drivers of *C. imicola* distribution in Europe [41,42]. The distribution and abundance of *C. imicola* are likely directly constrained by its relatively poor tolerance to relatively low temperatures [43]. Instead, crop irrigation practices in arid zones are assumed to support the presence of *C. imicola* [44,45]. The favorability predictive map partially agrees with the map developed by Ben Hassine et al. in 2021 using the ENFA and Maxent models coupled with WorldClim data [45]. The presence of sheep as a risk factor for *C. imicola* distribution was mentioned above [45]. In the second step, the favorability of encountering *C. imicola* (*C. imicola_F*) was used as an additional variable for the EHDV circulation prediction map. We did not force the model to use the potential distribution of *C. imicola* (*C. imicola_F* variable) on the GLM model but allowed the model to analyze and include it along with the other variables if it was statistically related to the 2021 outbreaks. Most remarkably, the inclusion of the vector variable (*C. imicola_F*) was the most influential variable on the distribution of infection with EHDV cases in Tunisia. Furthermore, these areas exhibited high variability in both day-night temperature (lst_dn_diff; 12–16 °C) and vegetation (NVDI_diff; 0.12–0.14). The ensemble model highlighted a large portion of central and north-western Tunisia (excluding regions with dense vegetation). Zones located inland demonstrated heightened variability, which was more favorable for infection with EHDV cases. These areas also include some of the most highly irrigated areas in Tunisia. Several coastal regions (Sahel and low steppes) and some regions in the Cap Bon peninsula in the far north-eastern region of Tunisia have been identified as suitable for EHDV circulation. In southern Tunisia, different areas located principally in oases have been identified as potentially suitable for EHDV circulation. Both the *C. imicola* and the EHDV risk models showed high sensitivity and acceptable specificity. Our ensemble models performed well, indicating a clear ability to distinguish between suitable and unsuitable habitat. Indeed, we assessed the ability of the 2021 infection with EHDV risk model to predict cases occurred in 2022.

These results suggest that *C. imicola* could be a potential vector of EHDV in Tunisia. Indeed, many cases of BTV–EHDV co-infections were confirmed in Tunisia in 2021 (23/161; 14.2%), suggesting that these two viruses can share the same epidemiological ecosystem. The co-circulation of BTV and EHDV has been reported in several countries and regions [46,47,48]. A significant association between cattle with BTV infections and the seroprevalence of EHDV was observed on cattle farms in China [19]. BTV seropositivity could therefore serve as a surrogate marker for the spread of EHDV. However, co-infection with different serotypes of EHDV and BTV increases the risk of potential genomic reassortment and is likely to pose a significant threat to cattle [19]. In fact, EHDV was isolated only from *C. kingi* and *C. oxystoma* pools in the oases of Tozeur, where *C. imicola* was not identified [13]. Dead deer were confirmed to be EHDV positive in the forest of Ghardimaou in the north-western Tunisia in 2021 and 2022 [13], a region not identified as suitable for EHDV circulation in this study, suggesting the possibility of intervention by other competent vectors. A recent survey in Italy showed that *C. obsoletus/scoticus* parous females have been found positive to EHDV-8 serotype. In Tunisia, the distribution of the species of *C. obsoletus* complex is limited to some sites in the northern part of country characterized by an ecosystem similar to that found in the southern part of Europe [9]. Other studies have demonstrated that there are considerable differences in the distributions and risk factors for these two viruses [49,50]. Boyer et al. (2010) found that EHDV seropositivity was associated with patches of forest, whereas BTV was not [49].

To create risk maps from a regression model, it is necessary to have spatial data layers for all covariates in the model. The use of other covariates like wind [50], drought severity [51,52,53], animal movement, and variables related to herd management or individual animal level factors may help to further improve predictions. In the case of wind, this variable can influence the distribution of *C. imicola* at large scales [43], and can also affect its propagation [52]. The translocation of *C. imicola* can lead to the spread of viruses associated with the vector to new places, as has occurred with BTV between Mediterranean islands [54]. In the case of EHDV, serotype 8 may have spread from Tunisia to Italy (Sardinia and Sicily) through *C. imicola* transported by wind currents that go from the African to the European continent. The EHDV risk map locates the areas of greatest risk in the north-east of the country, which in turn are crossed by strong wind currents (Figure 4). Strong winds may have carried infected *C. imicola* from Tunisia to the Mediterranean islands, spreading the disease and producing the first cases of EHD due to serotype 8 infection in Europe [14]. Since the first cases of this serotype in Tunisia in 2021, the virus may have persisted throughout the year. The following year, in 2022, as it was already widely distributed throughout the country, the virus could have spread to the Italian islands. In Figure 4, south winds blowing towards Sardinia (dated 23 October 2022) are shown, where infected vectors could potentially reach the coast of the island. Similarly, 10 days earlier, on October 13th, west winds blowing towards Sicily may explain the cases detected in this Italian island, which is closer to Tunisia (Ventusky 2023: https://www.ventusky.com/?p=35.1;7.1;5&l=wind-10m&t=20221013/1200, last accessed on 16 October 2023).

After spreading countrywide, endemization of EHD in the country is assumed to be quite probable [55]. Surveillance of this disease is recommended, and a clear control strategy should be defined [56]. This study defines EHDV potential distribution infection so that prioritized regions can then be indicated for disease/vector surveillance in Tunisia. Although, to date, there is no effective measure against EHDV-8 by chemical vector control, and there are no vaccines against EHDV-8 on which an efficientcontrol strategy is developed and implemented, the risk maps elaborated could be useful to implement other control measures such asanimal movement control, physical vector control, and use of an autogenous vaccine.

Future attempts to study the spatial distribution of BTV and EHDV seroprevalence, in combination with entomological studies, could help to understand the epidemiological difference between these two diseases. Furthermore, given the ease with which infected vectors can spread over long distances, these studies can be very useful to anticipate transboundary outbreaks.

## Figures and Tables

**Figure 1 viruses-16-00362-f001:**
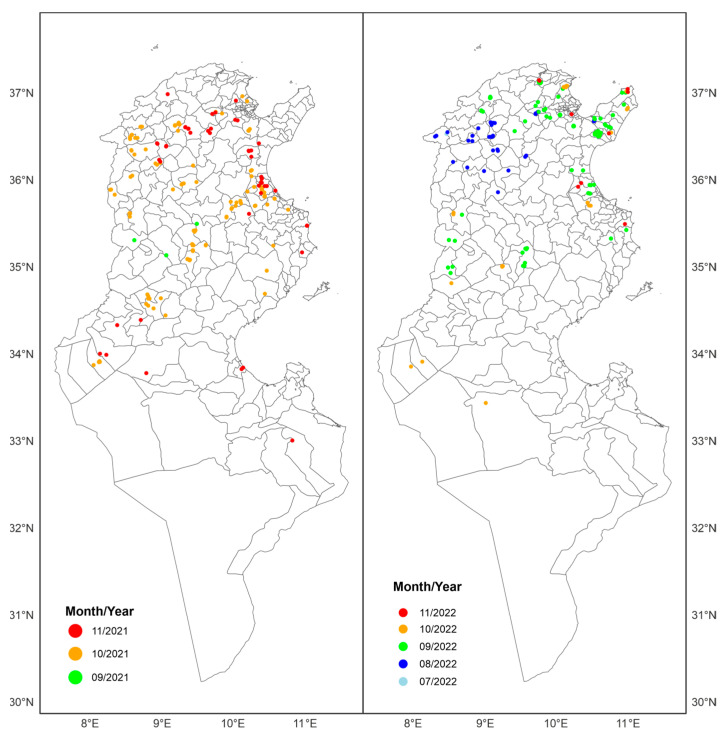
Spatiotemporal evolution of EHDV cases in livestock in the years 2021 and 2022, by months.

**Figure 2 viruses-16-00362-f002:**
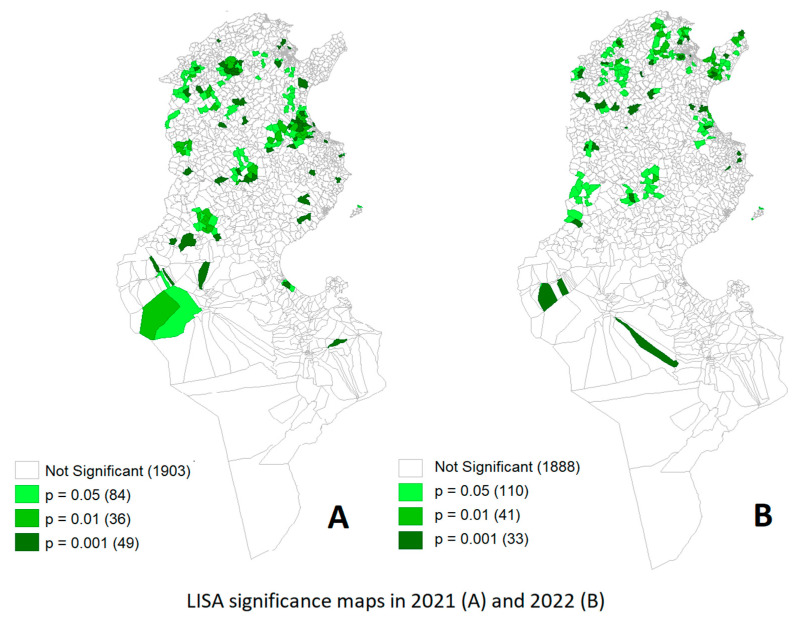
Clusters of spatial repartitions of positive EHDV herds by Imada (Third-level administrative division) in 2021 (**A**) and 2022 (**B**) identified by the LISA statistic.

**Figure 3 viruses-16-00362-f003:**
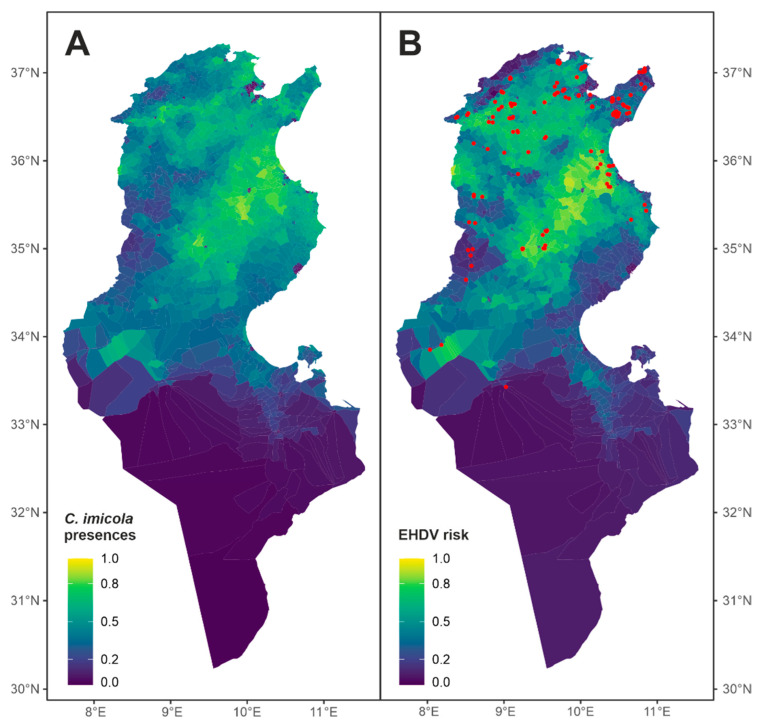
Environmental model of the presence of *Culicoides imicola* (**A**) and environmental risk model of EHDV cases in Tunisia (**B**). Red points in B represent the EDHV cases of the next year (2022).

**Figure 4 viruses-16-00362-f004:**
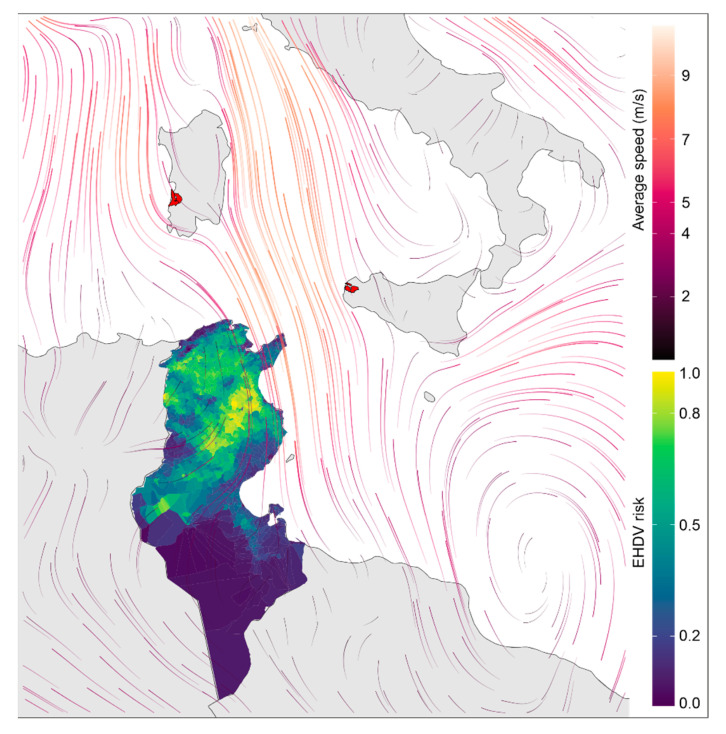
Environmental risk model of infection with EHDV cases in Tunisia together with cases due to EHDV serotype 8 infection in Italy (red polygons. Source: [14]). Wind speed on 23 October 2022, two days before cases in Italy. (Wind speed source: Climate Change Service, ERA5 hourly data on single levels from 1940 to present).

**Table 1 viruses-16-00362-t001:** Variables, and their factors, used to model *Culicoides imicola* and epizootic hemorrhagic disease cases’ probability of presence and favorability.

Factor	Variable	Code
Topography	Altitude ^1^	alt
	Slope ^1^	slope
Climatic	Day land surface temperature mean (2021) ^2^	dlst_mean
	Night land surface temperature mean (2021) ^2^	nlst_mean
	Land surface temperature day-night mean (2021) ^2^	lst_dn_mean
	Land surface temperature difference Day-Night (2021) ^2^	lst_dn_diff
Water availability	Distance to rivers ^3^	dist_river
	Percentage of areas equipped for irrigation ^4^	irrig
Livestock	Cattledensity ^5^	cattle
	Sheepdensity ^5^	sheep
Anthropic	Population density ^6^	dens_pop
Vegetation	Normalized Difference Vegetation Index mean (2021) ^7^	NDVI_mean
	Normalized Difference Vegetation Index minimum (2021) ^7^	NDVI_min
	Normalized Difference Vegetation Index maximum (2021) ^7^	NDVI_max
	Normalized Difference Vegetation Index difference between min and max (2021) ^7^	NDVI_diff

^1^ Global 30-Arc-Second Elevation Data Set for the World, developed by the United States Geological Survey (USGS) (http://eros.usgs.gov, last accessed on 6 September 2023). ^2^ MOD11A2 NASA product (1 km spatial resolution, temporal resolution 8 days) for the year 2021. Data were downloaded from the Land Processes Distributed Active Archive Center (LP DAAC) service at NASA website, (http://e4ftl01.cr.usgs.gov, last accessed on 16 September 2023). ^3^ Global Drainage Basin Database GDBD. Released Version 1.0: 29 May 2007 (http://www.cger.nies.go.jp/db/gdbd/gdbd_index_e.html, last accessed on 6 September 2023).^4^ Global Map of Irrigation Areas (version 4.0.1) around the year 2000 (http://www.fao.org/nr/water, last accessed on 17 October 2023).^5^ The agricultural map of Tunisia (National technical report, confidential data). ^6^ Landscan 2000 Global Population Database. Resolution 1 km × 1 km (https://landscan.ornl.gov, last accessed on 17 October 2023). ^7^ Global Agricultural Monitoring System (http://glam1.gsfc.nasa.gov/, last accessed on 11 December 2021).

**Table 2 viruses-16-00362-t002:** Explanatory variables included in the *Culicoides imicola* (up) and epizootic hemorrhagic disease virus models (down). Estimate is the coefficient that multiplies the variable values in the logit of the multivariate logistic regression. The Wald parameter quantifies the relevance of the variable in the model.

*Culicoides imicola* Model			
	Estimate	Wald	Significance
Intercept	−9.878	2.428 × 10^1^	8.338 × 10^−7^
Sheep	1.367 × 10^−4^	3.100	7.828 × 10^−2^
Irrigation	1.365	5.889	1.523 × 10^−2^
Dens_pob	−1.096 × 10^−3^	5.335	2.090 × 10^−2^
NLST_mean	4.143 × 10^−1^	1.014 × 10^1^	1.451 × 10^−3^
Dist_river	−9.285 × 10^−6^	4.005	4.538 × 10^−2^
**Epizootic hemorrhagic disease virus Model**			
	Estimate	Wald	Significance
Intercept	−1.594 × 10^1^	2.471 × 10^1^	6.679 × 10^−7^
*C. imicola_F*	4.124	2.974 × 10^1^	4.932 × 10^−8^
lst_dn_diff	3.567 × 10^−1^	1.898 × 10^1^	1.318 × 10^−5^
Slope	−3.698 × 10^−1^	1.327 × 10^1^	2.693 × 10^−4^
NDVI_diff	3.807	8.927	2.810 × 10^−3^
nlst_mean	−2.725 × 10^−1^	3.855	4.958 × 10^−2^

**Table 3 viruses-16-00362-t003:** Classification and discrimination capacities of the *Culicoidesimicola* and EHDV risk models.

	*Culicoides imicola* Model	EHDV Risk Model
Sensitivity	0.739	0.711
Specificity	0.607	0.637
Under-prediction Rate	0.010	0.023
Over-prediction Rate	0.957	0.907
Kappa	0.039	0.084
CCR	0.610	0.640
TSS	0.346	0.348
AUC	0.746	0.771

## Data Availability

The data presented in this study are available upon request to the corresponding author.

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
