# Peer review of "Epidemiological Analyses of the First Incursion of the Epizootic Hemorrhagic Disease Virus Serotype 8 in Tunisia, 2021–2022"

_viruses, 2024, doi:10.3390/v16030362_

Round 1
Reviewer 1 Report
Comments and Suggestions for Authors
The authors submitted the manuscript entitled "Epidemiological Analysis and Eco-climatic Factors of Epizootic Hemorrhagic Disease Outbreaks in Tunisia: Implications for Disease Spread" for publication in Viruses. The study focuses on the analysis of the 2021-2022 outbreaks of epizootic hemorrhagic disease (EHD) in Tunisia.
In this study, the authors employed generalized linear models (GLMs) to analyze the epidemiological situation of the EHD outbreaks in Tunisia. They examined the spatiotemporal distribution of the disease and attempted to elucidate the eco-climatic factors associated with its occurrence.
This manuscript highlights important implications for neighboring countries in Africa and Europe.
General comments:
The paper is overall quite readable. Several sentences should be rewritten and be improved (please see specific comments).
My main concern is about the inclusion of C.imicola_F in the EHDV model. I am no spatial-temporal analysis specialist, but it seems to me that when the same variables are used to assess both EHDV and C.imicola presence, the latter being the main vector of the disease, it is indeed expected that it is a main driver to detect the disease in livestock. Maybe I did not get this right, and if it is the case, it should nevertheless be clarified.
Specific comments:
Abstract:
L21: Culicoides should be italicized.
L23: Please change “subsequently” to another adverb as the spread from Tunisia to Italy, although probable, remains to be proven.
Introduction:
L38: Orbivirus should be italicized.
L42: add space between deer and [2]
L74: change “patterns of BTV” to “patterns as BTV”
Material & methods:
L139-151: Footnotes seem to be spaced too much and should be written in a smaller font.
Discussion:
L250: Please rephrase “like that of other…” as it does not sound as proper English
L285: How is it “most remarkably” ? It seems to me that the presence of the main vector should be a significant risk factor for the disease and quite expected.
L288: “That’s, the ensemble…” please rephrase
L303-306: Please change or delete this sentence. You cannot state in the M&M that you constructed a model based on the primary known EHDV vector and then in the discussion state that your results support the same vector to be a potential EHDV vector, that’s circular reasoning.
Comments on the Quality of English Language
The paper is overall quite readable. Several sentences should be rewritten and be improved (please see specific comments).
Author Response
The authors submitted the manuscript entitled "Epidemiological Analysis and Eco-climatic Factors of Epizootic Hemorrhagic Disease Outbreaks in Tunisia: Implications for Disease Spread" for publication in Viruses. The study focuses on the analysis of the 2021-2022 outbreaks of epizootic hemorrhagic disease (EHD) in Tunisia.
In this study, the authors employed generalized linear models (GLMs) to analyze the epidemiological situation of the EHD outbreaks in Tunisia. They examined the spatiotemporal distribution of the disease and attempted to elucidate the eco-climatic factors associated with its occurrence.
This manuscript highlights important implications for neighboring countries in Africa and Europe.
General comments:
The paper is overall quite readable. Several sentences should be rewritten and be improved (please see specific comments).
My main concern is about the inclusion of C.imicola_F in the EHDV model. I am no spatial-temporal analysis specialist, but it seems to me that when the same variables are used to assess both EHDV and C.imicola presence, the latter being the main vector of the disease, it is indeed expected that it is a main driver to detect the disease in livestock. Maybe I did not get this right, and if it is the case, it should nevertheless be clarified.
Response:
As the reviewer suggests, since C. imicola is the main vector of the disease, it is expected that the vector variable would be a key factor in detecting EHD cases, which is why we included it. Different models were developed before arriving at the current model. Models were developed using only disease cases, models were also developed considering points where the presence of C. imicola was confirmed, and finally a model was created considering the potential distribution of the vector (variable C.imicola_F). The models were evaluated, and the latter proved to be the best descriptive of the disease and also the best predictive model. The C.imicola_F variable includes areas environmentally favorable for the vector's presence, which is helpful in estimating areas where the vector might be and where presence points may be underestimated. In fact, when the model was run using the distribution of C. imicola instead of C.imicola_F, the model dismissed its distribution as an explanatory variable, indicating that the explanatory power of C.imicola_F goes beyond that of the distribution of C. imicola. We have rewritten the text in the results section to clarify this (lines 112).
We can add in the Methodology section, following to line 112 (after '(64 presence points)'): 'The C.imicola_F variable goes beyond the vector's own distribution, which may be greatly underestimated in the country, as it reflects environmentally favorable areas for the vector in Tunisia. This variable will allow the incorporation of information about the potential vector's distribution into the EHDV model.'
Specific comments:
Abstract:
L21: Culicoides should be italicized.
Response: your suggestion has been considered
L23: Please change “subsequently” to another adverb as the spread from Tunisia to Italy, although probable, remains to be proven.
Response : many thanks for your suggestion. You are absolutely right.
We propose to rephrase the text as follow : “The disease was detected later in the south of Italy, in Spain, in Portugal and, more recently, in France, where it caused severe infections in cattle”.
Introduction:
L38: Orbivirus should be italicized.
Response: your suggestion has been considered
L42: add space between deer and [2]
Response: your suggestion has been considered
L74: change “patterns of BTV” to “patterns as BTV”
Response: your suggestion has been considered
Material & methods:
L139-151: Footnotes seem to be spaced too much and should be written in a smaller font.
Response: your suggestion has been considered
Discussion:
L250: Please rephrase “like that of other…” as it does not sound as proper English.
Response: completely agree with your comment
The sentence has been changed: “As with other vector-borne diseases (VBDs), the distribution of EHD largely depends on the environmental factors that determine the abundance of the arthropod vector”.
L285: How is it “most remarkably” ? It seems to me that the presence of the main vector should be a significant risk factor for the disease and quite expected.
Response: We appreciate your comment as it has a major reformulation in the Materials and Methods section : C.imicola is considered as a possible main vector of the EHDV transmission not as the “main vector”. So, its inclusion as a possible risk factor is justified.
- We have no clear idea about the main vector of EHDV in Tunisia. In our last scientific paper “Thabet S, Sghaier S, Ben Hassine T, Slama D, Ben Osmane R, Ben Omrane R, Mouelhi W, Spedicato M, Leone A, Teodori L, Curini V, Othmani M, Berjaoui S, Ripà P, Orabi M, Mohamed BB, Sayadi A, Slama SB, Marcacci M, Savini G, Lorusso A, Hammami S. Characterization of Epizootic Hemorrhagic Disease Virus Serotype 8 in Naturally Infected Barbary Deer (Cervus elaphus barbarus) and Culicoides (Diptera: Ceratopogonidae) in Tunisia. Viruses. 2023 Jul 18;15(7):1567. doi: 10.3390/v15071567. PMID: 37515253; PMCID: PMC10383031, we discussed the possibility that other species of Culicoides, such as C.kingi and C.oxystoma can be vectors of EHDV in Tunisia .
- Other species have been proven to be competent vectors like C. Obsoletus in Europe.
So, the identification that EHDV repartition in Tunisia is related to the distribution of C.imicola is not quite expected for us.
We propose to rephrase the text as follow line 110 :”First, we constructed a distribution model for the possible primary vector responsible for transmitting the EHDV; C.imicola using occurrence points collected between 2017 and 2020 in Tunisia (64 presence points).”
L288: “That’s, the ensemble…” please rephrase
Response: Ok it’s corrected as you recommended
L303-306: Please change or delete this sentence. You cannot state in the M&M that you constructed a model based on the primary known EHDV vector and then in the discussion state that your results support the same vector to be a potential EHDV vector, that’s circular reasoning.
Response : We think that with the changes made below we can now accept this conclusion.
Reviewer 2 Report
Comments and Suggestions for Authors
Dear Authors
The study from a scientific point of view seems to be well done and presents good results, from where to derive valid conclusions. The issues raised in the article are consistent with the profile of the Journal. Abstract includes introductory statement that outlines the background and significance of the study. A well-documented introduction is presented, and the manuscript is well written. Methods are sufficient explained to replicate the research. The interpretation of the results is correct.
Minor revisions
Line 22,59 - Replace symptoms. with clinical signs
Line 38-39 genus Orbivirus in italic and the Family without italic
C. imicola is writed in 2 different ways: C. imicola and C.imicola. Plase correct in the texto to C. imicola.
Do not repeat results in discussion.
Author Response
The study from a scientific point of view seems to be well done and presents good results, from where to derive valid conclusions. The issues raised in the article are consistent with the profile of the Journal. Abstract includes introductory statement that outlines the background and significance of the study. A well-documented introduction is presented, and the manuscript is well written. Methods are sufficient explained to replicate the research. The interpretation of the results is correct.
Minor revisions
Line 22,59 - Replace symptoms. with clinical signs
Response: your suggestion has been considered
Line 38-39 genus Orbivirus in italic and the Family without italic
Response: your suggestion has been considered
- imicola is written in 2 different ways: C. imicola and C.imicola. Please correct in the text to C. imicola.
Response: your suggestion has been considered
Do not repeat results in discussion.
Response: your suggestion has been considered. Results are suppressed from line 463 to line 470
“Both the C. imicola and the EHDV risk models showed high sensitivity and acceptable specificity. Our ensemble models performed well indicating a clear ability to distinguish between suitable and unsuitable habitat. Indeed, we assessed the ability of the 2021 EHD risk model to predict cases occurred in 2022”.
Reviewer 3 Report
Comments and Suggestions for Authors
General comments
This paper describes the relationship between eco-climatic factors and the spread of EHDV based on data from Tunisia in 2021-2022, when bovine clinical cases occurred by EHDV-8 infection. In arboviral diseases of livestock animals, since various environmental factors as well as characteristics of the virus and immune status in susceptible animals are related to disease prevalence, it is not easy to identify environmental factors strongly associated with disease outbreaks. Therefore, it is interesting that the authors conducted an epidemiological analysis of several factors that could be related to the occurrence of EHD and showed factors that are strongly associated with the occurrence of the disease. Such studies may be useful in understanding the spread of arboviruses, and they may lead to epidemic forecasting in the future.
One point that needs to be corrected throughout the manuscript is the correct use of the terms EHD and EHDV. EHD is the name of the disease and EHDV is its pathogen, but there are many parts throughout the manuscript, including in the title and abstract, where the two terms EHD and EHDV are not used appropriately. For example, the title should be "… epizootic hemorrhagic disease virus serotype 8" and line 20 of the abstract should be "EHDV is transmitted…" in the abstract. The wording must be corrected.
Specific comments
Abstract, lines 33-34
The authors state that risk mapping could be useful for disease control and prevention strategies. However, there is no effective measures against EHD by vector control, and there are or vaccines against EHDV-8. So, I wonder how the risk mapping can be used for the disease control and prevention strategies. The authors should describe more specifically about that.
Materials and methods, lines 88-90
The authors mentioned that only the cases confirmed by the neutralization test and real-time RT-PCR were included in this analysis. However, EHDV infection often results in subclinical in cattle, and the period during which anti-EHDV antibodies and EHDV RNA are detected in infected cattle can be several months. So, I wonder how accurate the diagnosis was. There are many causes for fever, anorexia, dysphagia, etc. in cattle, so there may have been affected cattle with similar clinical symptoms with that of EHD but caused by other than EHDV infection. How did you differentiate them? I believe that information about the differentiation should be presented. If the purpose of this study is simply to analyze the spread of EHDV-8, there is no problem, but if the purpose is to analyze the spread of disease, the differentiation should also be explained. The authors should clearly indicate whether they analyzed the spread of EHDV-8 or the spread of EHD.
Line 94
When the word “Imada” is first mentioned, it should be stated that it is the smallest administrative unit in Tunisia.
Figure 1
Shouldn't there be a larger variation in the color intensity of the dots? I find it hard to tell the difference in color.
Figure 4
The possibility of EHDV-8 spread from Tunisia to Sardinia can be understood from the model shown in Figure 4. However, the possibility of spread to Sicily is not at all clear from this figure, nor is it mentioned in the text. The authors’ interpretation should be carefully explained in the manuscript.
Author Response
General comments
This paper describes the relationship between eco-climatic factors and the spread of EHDV based on data from Tunisia in 2021-2022, when bovine clinical cases occurred by EHDV-8 infection. In arboviral diseases of livestock animals, since various environmental factors as well as characteristics of the virus and immune status in susceptible animals are related to disease prevalence, it is not easy to identify environmental factors strongly associated with disease outbreaks. Therefore, it is interesting that the authors conducted an epidemiological analysis of several factors that could be related to the occurrence of EHD and showed factors that are strongly associated with the occurrence of the disease. Such studies may be useful in understanding the spread of arboviruses, and they may lead to epidemic forecasting in the future.
One point that needs to be corrected throughout the manuscript is the correct use of the terms EHD and EHDV. EHD is the name of the disease and EHDV is its pathogen, but there are many parts throughout the manuscript, including in the title and abstract, where the two terms EHD and EHDV are not used appropriately. For example, the title should be "… epizootic hemorrhagic disease virus serotype 8" and line 20 of the abstract should be "EHDV is transmitted…" in the abstract. The wording must be corrected.
Response: completely agree with your suggestion. As recommended the use of EHD and EHDV is corrected in many parts of the manuscript:
- The title is revised : “Epidemiological analyses of the first incursion of the epizootic hemorrhagic disease virus serotype 8 in Tunisia, 2021-2022”
- EHD is replaced by EHDV in : Line 21, Line 89, Line 147 ,Line 295, Line 300, Line 303, line 310 and 311 “ Figure 3”, Line 335, line 336, line 339 ,Line 348 “Figure 4”,Line 358, Line 359,Line 370, Line 374, Line 396, Line 401, Line 408 and Line 449
Specific comments
Abstract, lines 33-34
The authors state that risk mapping could be useful for disease control and prevention strategies. However, there is no effective measures against EHD by vector control, and there are or vaccines against EHDV-8. So, I wonder how the risk mapping can be used for the disease control and prevention strategies. The authors should describe more specifically about that.
Response: You are absolutely right. There is no effective measures against EHDV-8 by vector control, and there are no vaccines against EHDV-8 on which an efficient control strategy could be implemented. EHD may be difficult to control or eradicate once established, especially in a context of uncontrollable variables such as climatic and geographical factors including the abundance of suitable EHDV competent vectors. However, disease control and prevention strategies for EHD-8 include, in addition to the vaccination, which would be the most efficient if it existed, other means of control such movement control, vector control by physical means. However, for some serotypes other than EHDV-8,an autogenous vaccine can be used in certain animal species. These experiences could be tempted for EHDV-8.
Risk maps play a role in fortifying surveillance systems by pinpointing regions susceptible to disease transmission. This, in turn, streamlines the process of early outbreak detection.
We propose to add this paragraph in the discussion section: lines 457-461
“Although, to date, There is no effective measures against EHDV-8 by chemical vector control, and there are no vaccines against EHDV-8 on which an efficient control strategy developed and implemented, the risk maps elaborated could be useful to implement other control measures such as animal movement control, physical vector control and use of an autogenous vaccine”
Materials and methods, lines 88-90
The authors mentioned that only the cases confirmed by the neutralization test and real-time RT-PCR were included in this analysis. However, EHDV infection often results in subclinical in cattle, and the period during which anti-EHDV antibodies and EHDV RNA are detected in infected cattle can be several months. So, I wonder how accurate the diagnosis was. There are many causes for fever, anorexia, dysphagia, etc. in cattle, so there may have been affected cattle with similar clinical symptoms with that of EHD but caused by other than EHDV infection. How did you differentiate them? I believe that information about the differentiation should be presented. If the purpose of this study is simply to analyze the spread of EHDV-8, there is no problem, but if the purpose is to analyze the spread of disease, the differentiation should also be explained. The authors should clearly indicate whether they analyzed the spread of EHDV-8 or the spread of EHD.
Response: You are absolutely right. Clinical signs of EHD in cattle are similar to those of other diseases such BT, bovine viral diarrhoea/mucosal disease, infectious bovinerhinotracheitis, vesicular stomatitis, malignant catarrhal fever and bovine ephemeral fever. Definitive diagnosis of EHDV infection therefore requires the use of specific laboratory tests such sero-neutralisation and molecular techniques. Thus, we were not tempted to differentiate EHD from these diseases based on clinical signs. We rather opted to confirm the infection with EHDV. So, it would be more accurate to say the objective was to analyse the spread of the infection,as you mentioned,rather than the disease.
The sentence “The spatiotemporal distribution of the disease is described” line 100 is rephrasing as “the spatiotemporal distribution of infection with EHDV is described”.
Line 490: The sentence “ this study defines EHDV potential distribution” is rephrasing as
“This study defines EHDV potential distribution infection…”
Inappropriate use of EHD is corrected and replaced by EHDV in all parts of the manuscript.
Line 94
When the word “Imada” is first mentioned, it should be stated that it is the smallest administrative unit in Tunisia.
Response: your suggestion has been considered
Figure 1
Shouldn't there be a larger variation in the color intensity of the dots? I find it hard to tell the difference in color.
Response: I am attaching a new version of the Figure 1. in this new Fig. 1 the variation in the color intensity of the dots is larger, so reviewer can tell the difference in color easier.
Figure 4
The possibility of EHDV-8 spread from Tunisia to Sardinia can be understood from the model shown in Figure 4. However, the possibility of spread to Sicily is not at all clear from this figure, nor is it mentioned in the text. The authors’ interpretation should be carefully explained in the manuscript.
Response :Thank you for your suggestion: We propose to add this text, in line 446, after "... Italian islands". "In Figure 4, south winds blowing towards Sardinia (dated 23/10/2022) are shown, where infected vectors could potentially reach the coast of the island. Similarly, 10 days earlier, on October 13th, west winds blowing towards Sicily may explain the cases detected on this Italian island, which is closer to Tunisia (Ventusky 2023: https://www.ventusky.com/?p=35.1;7.1;5&l=wind-10m&t=20221013/1200)